# Novel Non-Immunologic Agents for Relapsed and Refractory Multiple Myeloma: A Review Article

**DOI:** 10.3390/cancers13205210

**Published:** 2021-10-18

**Authors:** Arthur Bobin, Cécile Gruchet, Stéphanie Guidez, Hélène Gardeney, Laly Nsiala Makunza, Mathilde Vonfeld, Anthony Lévy, Laura Cailly, Florence Sabirou, Thomas Systchenko, Niels Moya, Xavier Leleu

**Affiliations:** Department of Hematology, CIC 1402, University Hospital, 86000 Poitiers, France; cecile.gruchet@chu-poitiers.fr (C.G.); stephanie.guidez@chu-poitiers.fr (S.G.); helene.gardeney@chu-poitiers.fr (H.G.); laly.nsiala@chu-poitiers.fr (L.N.M.); mathilde.vonfeld@chu-poitiers.fr (M.V.); anthony.levy@chu-poitiers.fr (A.L.); laura.cailly@chu-poitiers.fr (L.C.); florence.sabirou@chu-poitiers.fr (F.S.); thomas.systchenko@chu-poitiers.fr (T.S.); niels.moya@chu-poitiers.fr (N.M.); xavier.leleu@chu-poitiers.fr (X.L.)

**Keywords:** multiple myeloma, non-immunologic agents, relapsed or refractory, IMiDs, proteasome inhibitors, XPO1 inhibitors, anti BCL-2, anti MCL-1, peptide–drug conjugates, HDAC inhibitors

## Abstract

**Simple Summary:**

Immunotherapy-based treatments have brought many new options in the treatment of multiple myeloma; still, the disease will inevitably relapse as it remains incurable. The development of non-immunologic drugs is therefore still needed, particularly for patients with advanced myeloma who become refractory to most of the drugs available, including immunotherapy. Non-immunologic agents have proven effective in the field for the past 20 years, notably with the advent of cytotoxic drugs and, subsequently, targeted therapy. In this review, we summarize the information currently available on novel generations of non-immunologic agents:proteasome inhibitors, immunomodulatory agents, anti-BCL-2/MCL-1, anti-XPO1, peptide–drug conjugates, and targeted drugs in early-stage development.

**Abstract:**

Novel treatments are needed to address the lack of options for patients with relapsed or refractory multiple myeloma. Even though immunotherapy-based treatments have revolutionized the field in recent years, offering new opportunities for patients, there is still no curative therapy. Thus, non-immunologic agents, which have proven effective for decades, are still central to the treatment of multiple myeloma, especially for advanced disease. Building on their efficacy in myeloma, the development of proteasome inhibitors and immunomodulatory drugs has been pursued, and has led to the emergence of a novel generation of agents (e.g., carfilzomib, ixazomib, pomalidomide). The use of alkylating agents is decreasing in most treatment regimens, but melflufen, a peptide-conjugated alkylator with a completely new mechanism of action, offers interesting opportunities. Moreover, with the identification of novel targets, new drug classes have entered the myeloma armamentarium, such as XPO1 inhibitors (selinexor), HDAC inhibitors (panobinostat), and anti-BCL-2 agents (venetoclax). New pathways are still being explored, especially the possibility of a mutation-driven strategy, as biomarkers and targeted treatments are increasing. Though multiple myeloma is still considered incurable, the treatment options are expanding and are progressively becoming more diverse, largely because of the continuous development of non-immunologic agents.

## 1. Introduction

The clinical course of multiple myeloma (MM) is characterized by a succession of remissions and relapses. Even though the past decade was marked by major therapeutic advances, thus increasing the survival of patients, MM is still considered to be incurable. As the disease progresses, the periods of remission progressively become shorter while the MM cells become more resistant, and eventually patients become refractory to all drugs available [1].

The advent of immunotherapy—particularly anti-CD38 monoclonal antibodies (mAbs)—was a major breakthrough for all patients with myeloma. However, in the near future, most MM patients will have been exposed early in the course of the disease to immunotherapeutic approaches, and particularly to anti-CD38 mAbs. Indeed, mAbs are now being used upfront for patients eligible and non-eligible for transplantation, or early in the relapse setting [2,3,4]. Moreover, at relapse there is a risk that the targeted antigen can diminish, as will its efficacy if deciding to re-treat with the same drug class [5,6]. Moreover, for the elderly, it is now acknowledged that the immune system can be physiologically dysfunctional; as such, immune-based treatments may not be used broadly at all lines of treatment [7]. Nevertheless, the immune-based treatment approach is evolving, notably with the emergence of active immunotherapy such as chimeric antigen receptor T (CAR-T) cells or bispecifics, but such attempts have failed to come closer to a cure thus far, as the remission obtained is only transient [8,9,10]. These modern immunotherapies are currently reserved for end-stage myeloma, but they will likely be used in early lines of treatment in the near future, hopefully with increased efficacy, and will also expose the targeted antigens relatively early in the course of myeloma without knowing whether retreatment with an immune-based treatment approach will be possible. Hence, even if immunotherapy is a step forward, the continuous search for new drugs is crucial, and novel non-immunologic agents are still needed in order to extend and diversify the treatment options that can bring clinical benefit in the early and late relapse settings. Additionally, there are some forms of myeloma—especially the most aggressive and bulky ones, or those previously treated with immunotherapy, which can develop mechanisms of resistance—that do not benefit the most from immune-derived treatments [11,12].

Immunomodulatory drugs (IMiDs) and proteasome inhibitors (PIs) have been the cornerstone of various drug regimens for several years now, largely overcoming traditional chemotherapy-based strategies. Recently, building on their efficacy, the new generation of IMiDs and PIs, that harbor different features, have proven to be suitable alternatives [13,14,15,16]. Exploiting well-known pathways is an effective approach but, most importantly, the understanding of MM’s pathophysiological mechanisms has led to the discovery of different therapeutic targets on plasma cells, and favored the development of completely new molecules, such as XPO1 inhibitors or HDAC inhibitors. Important changes have therefore occurred in MM treatment algorithms, especially at relapse, and novel non-immunologic agents are still playing a key role.

In this review, we discuss the most reliable and innovative options in the field of non-immunologic agents for relapsed or refractory multiple myeloma (RRMM).

## 2. The New Generations of Commonly Used Myeloma Drugs

### 2.1. The Search for the Ideal Proteasome Inhibitor

Proteasome inhibition forms part of the backbone of myeloma treatments. MM cells have high levels of proteasome activity, involved in the regulation of protein expression via the degradation of ubiquitylated proteins, which leads to the suppression of apoptotic pathways that contribute to cell survival. Bortezomib (V) was the first-in-class proteasome inhibitor, and was next followed by carfilzomib (K), the second-generation PI that has widened the treatment options for RRMM patients. Carfilzomib was evaluated at relapse in the two pivotal phase III studies ENDEAVOR (Kd vs. Vd) and ASPIRE (KRd vs. Rd), with results in favor of carfilzomib [13,14]. Even though carfilzomib is an interesting option, and can be used in bortezomib-refractory patients, it comes with new side effects that were not reported previously with the first PI. Despite a decreased rate of peripheral neuropathy (PN) compared to bortezomib—one of the major drawbacks of this drug—carfilzomib is associated with a risk of cardiac toxicity, especially arterial hypertension or heart failure, which might require special attention. Additionally, a decrease in renal function has been noted in some patients, which it is now recommended to monitor throughout the treatment course. Moreover, one could once have said that the main inconvenience of carfilzomib was its bi-weekly intravenous (I.V.) administration (27 mg/m^2^). However, the phase III A.R.R.O.W. trial demonstrated that the once-weekly 70 mg/m^2^ I.V. administration, along with dexamethasone, led to a longer progression free survival (PFS) (median PFS 11.2 vs. 7.6 months) with an acceptable safety profile [17]. The A.R.R.O.W. trial led to the approval of Kd with K 70 mg/m^2^ once per week. The following phase III A.R.R.O.W.2. (NCT03859427) will further investigate K weekly in association with lenalidomide and dexamethasone versus the approved regimen of KRD with K bi-weekly.

The next step in the development of PIs was the production of an oral agent so that their administration would be easier for patients. This happened with ixazomib (I), the first orally administered PI to be used in MM. Its approval for RRMM patients was based on the results of the phase III TOURMALINE-MM1 trial (IRd vs. Rd) [18]. The toxicities of ixazomib are mainly hematologic and gastrointestinal (GI) events, and the rate of PN is also relatively low. It is noteworthy that ixazomib can be used in patients exposed to bortezomib, but not in bortezomib- or carfilzomib-refractory patients. Furthermore, at relapse, carfilzomib is probably still a preferred option, as it will lead to a prolonged survival. Indeed, even though there is no head-to head comparison, median PFS with KRd in the ASPIRE trial was longer than that obtained with IRd in the TOURMALINE-MM1 trial: 26.3 months vs. 20.6 months, respectively. In addition, recently, the final analysis of TOURMALINE-MM1 demonstrated no overall survival (OS) benefit for IRd in RRMM in comparison with the control arm (median OS 53.6 months for IRd vs. 51.6 months for Rd, HR = 0.939, 95%CI = 0.784–1.125, *p* = 0.495). Ixazomib was also evaluated as a maintenance treatment (for 26 28-day cycles) after induction for non-transplant-eligible (NTE) patients in the placebo-controlled TOURMALINE-MM4 trial, showing a benefit in PFS (17.4 months vs. 9.4 months for the placebo arm, *p* < 0.001). Ixazomib is the first oral PI to demonstrate a significantly longer PFS as maintenance therapy in NTE patients. However, the control arm was likely suboptimal and, more importantly, these results will be expected to translate into an improved OS before ixazomib can find a place in this setting. Overall, ixazomib remains an interesting option, especially for elderly patients, because it can be integrated in all-oral treatment regimens.

Ideally, new proteasome inhibitors would be oral agents—or perhaps subcutaneous (S.C.)—for convenient administration and an acceptable quality of life for all patients. Most importantly, they should overcome bortezomib and carfilzomib resistance and, eventually, should have a tolerable safety profile that could allow their administration on a long-term basis, especially with reduced neurological and cardiological toxicity. The attempts to develop such drugs are still ongoing, as the two recent attempts with oprozomib and marizomib did not go as expected [19,20]. These next-generation oral PIs were promising at first, but oprozomib, though active in bortezomib-refractory patients, led to troubling gastrointestinal disorders—particularly gastrointestinal hemorrhage—and marizomib, while less toxic, was not so effective. New formulations of these drugs should be expected; until then, the search for a novel PI that could be incorporated into the myeloma armamentarium is still in progress.

### 2.2. Novel Immunomodulatory Agents: Are CELMoDs the Future of Myeloma?

As with proteasome inhibitors, immunomodulatory agents are highly successful in MM. IMiDs have both direct (binding to the cereblon E3 ubiquitin ligase complex) and indirect (increased expression of interleukin-2 from T-cells, leading to natural-killer- and antibody-mediated toxicity) mechanisms of action [21,22]. After thalidomide and lenalidomide, pomalidomide was the third IMiD to become part of MM treatments. Pomalidomide was initially evaluated for patients with RRMM in the phase III MM-003 trial with either low-dose dexamethasone or high-dose dexamethasone, and then in the phase III OPTIMISMM trial (Pd vs. Vd) [15,23]. Pomalidomide has become a relevant option for lenalidomide-refractory patients, who now represent a growing number of patients with RRMM, as lenalidomide is routinely used frontline or early in the treatment course. However, results with pomalidomide-based regimens are rarely superior to those obtained with lenalidomide-based treatments. In this context, there is a need for novel IMiDs with better efficacy and that would be active in lenalidomide- and pomalidomide-refractory patients.

These issues led to the development of the new IMiD-derived class of cereblon E3 ligase modulators (CELMoDs), which are also oral agents. CELMoDs have an increased affinity to cereblon compared to IMiDs and, thus, the degradation of IKZF1 (Ikaros) and IKZF3 (Aiolos)—transcription factors essentials for MM cell survival—is increased, eventually leading to a greater antiproliferative effect [24] (Figure 1). CELMoDs have shown promising results in preclinical models, and can overcome IMiD refractoriness; thus, they entered phase I/II trials for RRMM.

#### 2.2.1. CC-220 (Iberdomide)

First, CC-220 demonstrated a greater anti-myeloma activity than lenalidomide or pomalidomide in both sensitive and resistant MM cell lines. This led to the first-in-human multicohort phase I/II trial (NCT02773030) for RRMM (≥2 lines of treatments, including a lenalidomide/pomalidomide and a PI) to determine the maximal tolerated dose (MTD) of iberdomide alone or in combination. The MTD was not reached, and the recommended phase II dose (RP2D) was 1.6 mg from days 1–21, in a 28-day cycle. The study of CC-220 and dexamethasone showed an overall response rate (ORR) of 32.2% (35.3% in IMiD refractory) for all evaluable patients (*n* = 58, median of 5 prior lines) [25] (Table 1). For the triplet association of iberdomide, bortezomib, and dexamethasone (*n* = 23, median of 6 prior lines), the ORR was 60.9%. In this cohort, 78.3% of patients were IMiD-refractory and 65.2% were PI-refractory; response was independent of PI or IMiD refractoriness. Of note, in this study, iberdomide was also combined with daratumumab (D) (ORR = 42.3%). Several trials are ongoing with CC-220, notably for RRMM patients; the ICON trial (phase II, 2–4 prior lines of treatment) will evaluate the combination of iberdomide, low-dose cyclophosphamide (50 mg on days 1–28), and dexamethasone, while the phase III EXCALIBER trial will investigate the combination of iberdomide, daratumumab, and dexamethasone in comparison with DVd (EudraCT 020-000431-49) (Table 2).

#### 2.2.2. CC-92480

CC-92480 is a novel CELMoD agent that, as with CC-220, showed in vitro anti-myeloma activity, including in lenalidomide- and pomalidomide-resistant cell lines. CC-92480 was tested in a multicenter phase I trial (NCT03374085), in heavily pretreated patients (*n* = 66, median of prior treatments = 6). The MTD was 1.0 mg for the two different schedules of administration [26]. The ORR was 21% for all evaluable patients, and 48% at the therapeutic dose of 1.0 mg (Table 1). A phase I/II study is ongoing for RRMM and newly diagnosed MM to determine the RP2D of CC-92480 in combination with standard treatments (bortezomib, carfilzomib, anti-CD38 mAbs, etc.) (NCT03989414).

#### 2.2.3. Safety Profile

With CELMoDS the main AEs are hematological—principally myelosuppression. In the phase I/II study of CC-220 and dexamethasone, 72% of the patients experienced grade 3–4 AEs, including neutropenia (26%), thrombocytopenia (11%), and neuropathy (2%). There were no patients who experienced dose-limiting toxicities (DLTs) for doses from 0.3 to 1.1 mg, but one patient had grade 4 sepsis at 1.2 mg, while another had grade 3 pneumonia at 1.3 mg of CC-220. Regarding the combination with bortezomib and daratumumab, 65% and 78% of patients had grade 3–4 AEs, respectively. Again, the most frequent AEs were neutropenia, leukopenia, and anemia. This toxicity profile is relatively similar to that of CC-9280, for which the phase I trial reported 88% of patients with grade 3–4 AEs, including neutropenia (53%), infections (30%), anemia (29%), and thrombocytopenia (17%). Furthermore, 10 patients had DLTs, mostly due to neutropenia.

### 2.3. Melflufen, a Peptide–Drug Conjugate: The Evolution of Alkylating Agents

Even though alkylating gents are decreasingly used in the novel agent era, they have been a major step in the history of myeloma. Today, they still have not been replaced for transplant conditioning, and their use is warranted in some of the patients who will largely benefit from their efficacy over targeted therapies. However, their systemic toxicity will be limiting in most of the cases and, therefore, they can rarely be given long term. With the use of alkylating agents also comes the risk of secondary malignancies—a major issue for the future of the patients. Additionally, patients can experience hair loss if subject to prolonged exposure, which can be traumatizing for most of them. For these reasons, alkylating agents had to evolve in order to retain their effect on plasma cells while limiting their toxicity.

The development of melphalan flufenamide (melflufen) could address these concerns. Melflufen is a first-in class peptide-conjugated alkylator that is rapidly internalized into cells due to its high lipophilicity (Figure 1). Melflufen, as a prodrug, is metabolized by aminopeptidases—which are increased in MM cells [27]—and selectively releases an alkylating agent into tumor cells to exert its cytotoxic activity [28]. This approach is different from that of conventional alkylating agents, which have been used in MM since the 1960s, as it allows a higher intracellular concentration of the cytotoxic agent directly within the targeted cells. As a result, preclinical studies have shown that melflufen exhibits a higher cytotoxicity compared to melphalan. Moreover, it is likely to overcome melphalan resistance, as demonstrated in preclinical models [29].

Melflufen was recently approved (2021) by the US Food and Drug Administration (FDA) for patients with RRMM who had received four prior lines of treatments, including one PI, one IMiD, and one anti-CD38 mAb, based on the results of the phase II HORIZON trial [30]. The study, whose primary endpoint was the evaluation of ORR, enrolled 157 patients (median of five prior lines of treatment) refractory to pomalidomide and/or anti-CD38 mAb, who were treated with 40 mg of melflufen I.V. every 4 weeks and 40 mg of dexamethasone (20 mg if >75 y.o.) weekly. The ORR was 29%, median PFS was reported at 4.2 months, and median OS was 11.6 months, with a median follow-up of 14 months (Table 1). Interestingly, activity was also observed among triple-class-refractory (refractory to PIs, IMiDs, and anti-CD38 mAbs) patients, with an ORR of 26%, which is approximately similar to the result obtained for the all-treated population, as well as in patients with extramedullary disease (24%) and patients ≥75 years old (32%). Moreover, the subgroup analysis showed that melflufen was efficient in patients refractory to an alkylator in one previous line of therapy (ORR 28%), which is consistent with preclinical data and the different mechanism of action of melflufen. Grade ≥ 3 AEs occurred in 96% of the patients; most of them were hematologic events (neutropenia 79%, thrombocytopenia 76%, and anemia 43%), while pneumonia was the most common non-hematologic event (10%). Of note, gastrointestinal events were frequent (62%), but predominantly grade 1/2 (93%). A total of 42 patients (27%) experienced melflufen dose reductions because of treatment-emergent AEs (TEAEs)—principally thrombocytopenia (14%) and neutropenia (3%)—and 34 patients (22%) discontinued the treatment for a TEAE at least once, again for the same causes. With alkylator agents comes the risk of secondary myelodysplasia, which one patient developed after 17 cycles of melflufen (but in the context of multiple prior lines of alkylators).

The phase III randomized, multicenter, open-label, OCEAN trial (NCT03151811) is still ongoing, and is evaluating melflufen and dexamethasone versus pomalidomide and dexamethasone (Pd) for patients with RRMM with 2–4 prior lines of treatment—including a PI and lenalidomide—and refractory to the last line of therapy and lenalidomide (Table 2). The primary outcome is the measure of PFS. Though the study has not been completed, the preliminary results demonstrate that melflufen–dexamethasone is not inferior to Pd (*p* = 0.0640), and median PFS was 41% superior in the experimental arm. Similarly, the ORR was 32.1% for melflufen and 26.5% for pomalidomide.

Melflufen is also being tested in association with other MM agents, as in the phase I/II ANCHOR study (OP-104; NCT03481556), where melflufen is combined with either bortezomib or daratumumab for RRMM. The first results are showing promising results, with ORRs of 73% and 62% in the daratumumab and bortezomib groups, respectively [31]. This study will be followed by the phase III LIGHTHOUSE trial (NCT04649060) using melflufen–daratumumab–dex vs. daratumumab as a single agent (Table 2). 

It should be noted that, in July 2021, the FDA suspended the enrollment in all of the ongoing clinical trials evaluating melflufen as a result of the increased risk of death in the OCEAN trial. The safety of melflufen will have to be carefully discussed in line with this new information.

## 3. Targeting New Pathways in Multiple Myeloma

### 3.1. Anti-BCL-2/MCL-1: A Tailored Therapy

The members of the B-cell lymphoma 2 (BCL-2) protein family regulate the intrinsic apoptotic pathway, including proapoptotic (BAX, BAK, BIM, BAD) and antiapoptotic (BCL-2, MCL-1, BCL-XL, BCL-W) proteins [32] (Figure 1). In myeloma, the expression of antiapoptotic BCL-2 proteins is elevated, thus promoting MM cells’ survival, although the dependence on BCL-2 for cell survival is heterogeneous from one patient with MM to another. However, the expression of MCL-1 is increased in most MM cell lines [33], and represents a potential mechanism of resistance to BCL-2 inhibition. The subset of patients carrying the t(11;14) translocation (15–20% of patients), which comes with higher levels of BCL-2 but a lower expression of MCL-1/BCL-XL, will likely benefit the most from the anti BCL-2 approach [34]. Targeting the apoptotic pathways instead of focusing mainly on proliferative cells is an interesting approach in myeloma, as MM cells can be quiescent inside the bone marrow niche.

#### 3.1.1. BCL-2 Inhibitors

Venetoclax is an oral, potent, selective inhibitor of BCL-2 that has proven effective in various hematological malignancies—especially chronic lymphocytic leukemia (CLL) [35] or acute myeloid leukemia (AML) [36]. Recently, it has also demonstrated activity on MM cells, which led to its investigation in clinical trials. In myeloma, however, the dosage of venetoclax is different from what is commonly approved for CLL or AML (400 mg/day), at 800 mg daily. The phase III, randomized, double-blind, multicenter BELLINI trial included 291 patients with RRMM who had received 1–3 prior lines of therapy, and who were treated with either venetoclax, bortezomib, and dexamethasone, or bortezomib, dexamethasone and placebo [37]. The association with bortezomib was based on the demonstration that PIs can increase BCL-2 expression and downregulate MCL-1, therefore potentially enhancing venetoclax’s action. With a median follow-up of 18.7 months, the median PFS was 22.4 months in the venetoclax group compared to 11.5 months in the placebo group (*p* = 0.010). However, there was an increased risk of mortality in the experimental arm (14 treatment emergent deaths vs. 1), mainly due to disease progression and infections (8/14, 57.1%), although the rates of serious infections were comparable between the two arms. Most importantly, the subgroup analysis revealed that the higher rates of response were for the patients with the t(11;14) translocation; notably, the MRD 10^−5^ rate was 25% for this group, which compares favorably to the MRD negativity rates obtained with immunotherapy-based triplet combinations in other trials. Moreover, they also displayed the longest PFS (median PFS not reached). Furthermore, mortality in the t(11;14) or high-BCL-2-expression subgroups was similar between the two treatment groups. 

For that reason, ongoing venetoclax-based trials are mainly focusing on patients with the t(11;14) translocation, marking the first biomarker-driven approach in MM. Two studies of carfilzomib, venetoclax, and dexamethasone also support the role of venetoclax in t(11;14)-positive patients, as the ORR was 100% for them in both studies, while mixed responses were noted for t(11;14)-negative patients (ORR of 74% in one study, and no response in the other). The role of venetoclax will be further explored in the phase III CANOVA trial (NCT03539744), which is evaluating the combination of venetoclax and dexamethasone for RRMM patients positive for the t(11;14) translocation (Table 2). 

Of note, with the use of venetoclax comes the risk of developing acquired mutations and, therefore, resistance to the treatment, as has previously been shown in CLL (mutations in the BCL-2 family such as *BCL2* or *BAX*, as well as mutations involving *BRAF*, *CDKN2A/B*, *BTG1*). 

#### 3.1.2. MCL-1 Inhibitors

In addition to BCL-2 inhibition, targeting MCl-1 is an attractive possibility given that it is overexpressed in MM for most patients, and it is involved in venetoclax resistance. A dual inhibition of BCL-2 and MCL-1 could therefore be of major interest [38]. For now, various anti-MCL1 drugs are in early-phase development, including S64315 (also known as MIK665)—which demonstrated anti-MM activity in preclinical models, and is currently being evaluated in a phase I trial for RRMM (NCT02992483)—as well as AMG176, AMG397, and AZD5991, which are also being tested in phase I studies (NCT02675452, NCT03465540, and NCT03218683). However, these drugs should be considered with caution for now, as few data are available, and two trials (AMG176 and AMG397) were put on hold to evaluate signs of cardiac toxicity with AMG176.

### 3.2. Selective Inhibitors of Nuclear Export (SINE): A Completely New Family in Myeloma 

Exportin-1 (XPO1) is an oncoprotein that mediates the nuclear export and inactivation of several tumor-suppressor proteins. XPO1 is overexpressed in various cancers, including MM [39,40,41]. Selective inhibitors of nuclear export (SINEs) therefore force the nuclear localization and activation of tumor-suppressor proteins, resulting in apoptosis of malignant cells [42] (Figure 1). Selinexor, the first-in-class SINE, is already approved for some indications, and the second-generation eltanexor has also entered clinical trials. This completely new family in MM represents another option to treat patients—especially those who are refractory to IMiDs.

#### 3.2.1. Selinexor

Selinexor (80 mg twice weekly) was evaluated in the phase IIb single-arm, multicenter STORM trial in combination with dexamethasone (20 mg) for patients with RRMM who had been treated with lenalidomide, pomalidomide, bortezomib, carfilzomib, and daratumumab (penta-exposed), and were triple-class refractory [43]. The study enrolled 122 previously highly treated patients (median of previous regimens = 7) for whom the ORR—the primary endpoint—was 26%, median PFS was 3.7 months, and median OS was 8.6 months (Table 1). Furthermore, the ORR of the penta-refractory patients was 25%. 

Selinexor-based triplets are under evaluation for RRMM patients, and will potentially address the lack of treatments for IMiD-refractory patients. The phase III BOSTON trial was designed to analyze the efficacy of selinexor (at a different dosage: 100 mg weekly, 5-week cycles) combined with bortezomib (1.3 mg/m^2^ SC weekly, 5-week cycles) and dexamethasone, in comparison with bortezomib and dexamethasone alone (Vd). A total of 402 patients (median number of previous lines = 2) were randomized, and the primary endpoint was PFS; median PFS was significantly longer for the selinexor group (13.93 months) compared to Vd (9.3 months) (*p* = 0.0075) [44] (Table 2). The median time to the next MM treatment—an important datum in the relapse setting—was longer in the selinexor group (16.1 months vs. 10.8 months for Vd, *p* = 0.0012). Multiple selinexor-based combinations (10 combinations, 11 arms) are being evaluated in the phase I/II STOMP study (≥3 prior lines of therapy, including a PI and an IMiD); the preliminary results of the selinexor–carfilzomib–dex arm found an RP2D of 80 mg weekly, an ORR of 70.8%, and a median PFS that was not reached (median follow-up of 4.7 months) [45], while the selinexor–pomalidomide–dex arm found the RP2D at 60 mg weekly, the ORR at RP2D was 65%, and the median PFS for all patients was 10.4 months [46]. The results of the other arms of the STOMP study are awaited.

Concerns exist about the safety profile of selinexor. A retrospective pooled analysis of selinexor trials (STORM, STOMP, and BOSTON; *n* = 437) revealed high rates of non-hematologic events such as nausea (68%), decrease in appetite (53%), diarrhea (41%), and vomiting (37%) [47]. Nevertheless, these side effects were mainly grade 1–2, and occurred during the first weeks of treatment. As for hematologic AEs, they were principally thrombocytopenia (66%) and neutropenia (37%). However, tolerance to selinexor can be improved with careful management. The use of dexamethasone and prophylactic antiemetics is warranted due to the GI toxicity; as for the hematologic toxicity, selinexor should not be administered unless platelets are above 50 G/L, in order to avoid high-grade thrombocytopenia and, potentially, bleeding or hemorrhage. Treatment interruptions and platelet infusions can therefore be required. Interestingly, thrombocytopenia related to selinexor seems to be dose dependent, so dose reductions when resuming the treatment after a pause are important [48].

There is still a debate about the proper dosage of selinexor in order to maintain good efficacy and prevent toxicity. Indeed, in the STORM trial, the posology of selinexor was clearly higher than in BOSTON: 80 mg twice a week (i.e., 160 mg weekly) versus 100 mg weekly. One could argue that in a triplet combination a higher dosage of selinexor is not required, but there is currently no proof that selinexor can be as effective as a single agent at this weekly dosage. The STOMP study, with its evaluation of 10 different combinations at various dosages of selinexor, will probably help to clarify the appropriate scheme of administration of selinexor used in combination therapy.

#### 3.2.2. Eltanexor

Finally, another SINE—eltanexor—could find its way into MM treatments, potentially with fewer side effects. Eltanexor is a second-generation SINE that has reached human clinical trials—notably, a phase I/II study of 36 patients with a median of 7 prior therapies, where although it was relatively well tolerated the response rate was limited (ORR = 13%) [49]. More data on eltanexor are therefore awaited.

### 3.3. HDAC Inhibitors: Promises and Some Doubts

Histone deacetylases (HDAC) inhibitors target histone modifications that lead to different chromatin conformation and DNA transcription, thus regulating gene expression and cell survival [50]. In the context of MM, HDAC inhibitors reactivate silenced genes and, therefore, cause cell death. Two HDAC inhibitors have been principally used in MM: panobinostat—a pan-HDAC inhibitor—and ricolinostat—a HDAC6 inhibitor—though other agents are also available (vorinostat, romidepsin, etc.) [51]. Unfortunately, the efficacy of HDAC inhibitors as single agents is limited in MM [52]. Nevertheless, their use in combination—especially with PIs, because of their synergistic activity [53,54]—can represent an alternative option with a different mechanism of action for patients with RRMM.

#### 3.3.1. Pan-HDAC Inhibitors

The phase III PANORAMA-1 trial (*n* = 768) investigated panobinostat in combination with I.V. bortezomib and dexamethasone versus bortezomib and dexamethasone for patients who had received 1–3 prior lines of therapy. The primary endpoint was met, with a median PFS of 12 months for the panobinostat group and 8.1 for Vd (*p* = 0.0001) [55]. Then, the multicenter phase II PANORAMA-3 study was designed to evaluate S.C. bortezomib with panobinostat at different dosages (20 mg × 3/weeks; 20 mg × 2/w; 10 mg × 3/w) and dexamethasone [56]. In this trial, 248 patients (median of 2 prior lines of treatment; 68% of patients had previous bortezomib exposure) were enrolled, having received 1–4 previous treatments; ORR was the highest in the 20 mg three times a week (the approved schedule) and 20 mg twice weekly groups, at 62.2% and 65.1%, respectively, while the therapy was best tolerated in the 10 mg group. Based on these encouraging results with PIs, panobinostat was combined with carfilzomib in two phase I studies; ORR was reported at 67% and 57%, and median PFS at 7.7 months and 8 months in the two studies [57,58]. The treatment related toxicities were mainly of hematologic (especially thrombocytopenia) and gastrointestinal origin. Diarrhea was an important concern in the PANORMA-1 study, affecting 76% of patients (33% of grade 3/4 toxicities).

Combinations of panobinostat and IMiDs were also evaluated for RRMM patients, notably in a single-center phase II trial of panobinostat, lenalidomide, and dexamethasone (*n* = 27, median of 3 prior lines of therapy) [59]. The ORR was 41%, and median PFS was 7.1 months. Of note, 81% of the patients were lenalidomide-refractory. Interestingly, in comparison to the PANORAMA-1 study, there were no significant GI toxicities.

Panobinostat-based quadruplet combinations were also tested in RRMM patients, as in the panobinostat–VTD regimen (phase I/II, *n* = 57, ORR 91%) [60], and the panobinostat–VRD regimen (phase I, *n* = 20, ORR 44%) [61].

#### 3.3.2. HDAC6 Inhibitors

While pan-HDAC inhibitors are disappointing as single agents, the development of a more selective oral agent was warranted. This led to the arrival of ricolinostat—an HDAC6 inhibitor—which was investigated in patients with RRMM in hopes of less toxicity and improved efficacy. Ricolinostat was investigated in a multicenter phase I/II trial but, again, it was not clinically active as a single agent. However, the ORR for ricolinostat (at the RP2D of 160 mg daily) in combination with bortezomib was 37% (14% for bortezomib-refractory patients) [62]. As compared to data on non-selective HDAC inhibitors, there were less hematologic and GI toxicities. Ricolinostat was also combined with lenalidomide in a multicenter phase Ib trial where 38 patients were included. Similarly, the RP2D was 160 mg daily on days 1–21 (28 days cycle), and the ORR was reported at 55% [63]. Again, the treatment was well tolerated (diarrhea grade 1–2 in 39% of patients; grade 3 in 5%).

Overall, HDAC inhibitors are not widely implemented in clinical practice, probably due in part to their toxicity profile (panobinostat was approved with a warning for severe diarrhea and cardiac AEs). Additionally, panobinostat is only approved in association with I.V. bortezomib (and dexamethasone) on the basis of the PANORAMA-1 study—a combination that is more toxic than S.C. bortezomib, the new standard of care. However, HDAC inhibition remains an interesting mechanism of action in MM, with in vitro demonstration of its efficacy, although there is room for improvement, as targeting this sole pathway does not lead to the expected results in clinical practice. For example, it would be relevant to determine whether some selected patients could benefit more from HDAC inhibitors than others or, simply, there might be another approach to target histone modification, but it is yet to be discovered.

## 4. New Emerging Agents: Perspectives for the Future

### 4.1. Other Targeted Therapies

Several drugs are in early-stage development for MM, using innovative mechanisms of action. Currently, we cannot assert that these emerging agents will one day be investigated in phase III trials; however, they are worth mentioning, as they also allow us to better understand the complexity of MM and its microenvironment, and they still represent interesting approaches for the future.

#### 4.1.1. Dinaciclib

Dinaciclib is a selective inhibitor of cyclin-dependent kinases (CDK1, 2, 5, and 9), whose dysregulation is frequently found in MM. CDKs are involved in the regulation of the cell cycle and DNA repair. It was shown in vitro that inhibition of CDK5 could enhance the activity of PIs, thus making CDK inhibitors an interesting option for the treatment of MM. Dinaciclib was tested as single agent in a phase I/II study for patients with RRMM (≤5 prior lines of therapy, 27 evaluable patients); the ORR was 11% overall, with two patients achieving VGPR, and the median PFS across the entire study was 3.5 months [64]. The most common AEs were leukopenia, thrombocytopenia, gastrointestinal symptoms, and fatigue. Hopefully, the results will be improved with the use of dinaciclib in combination with other MM drugs; the results of the phase I study of dinaciclib, bortezomib, and dexamethasone are yet to be unveiled (NCT01711528). 

#### 4.1.2. Ruxolitinib

Targeting the Janus tyrosine kinase (JAK) pathway is effective in treating various hematological malignancies. The JAK pathway is involved in the activation of transcriptional factors that lead to cell proliferation and survival. After preclinical models showed that ruxolitinib—a JAK 1 and 2 inhibitor—was active on MM cells and could restore sensitivity to lenalidomide, a phase I trial (*n* = 28, median of 6 prior treatments) with ruxolitinib, lenalidomide, and corticosteroids was therefore designed for RRMM patients. The ORR was 38%, and all of the responding patients were lenalidomide-refractory [65]. A phase I/II study investigating ruxolitinib with carfilzomib is also ongoing (NCT 03773107).

#### 4.1.3. MAPK Inhibitors

Mutations of the mitogen-activated protein kinase (MAPK) pathway are common in MM (approximately 50% of newly diagnosed patients, and even more for RRMM), including driver mutations of KRAS, NRAS, and BRAF (2–4% of newly diagnosed patients and 8% of RRMM). Case reports of MM patients exhibiting BRAF V600 mutation showed successful treatment with BRAF inhibitors. There are several selective BRAF inhibitors: vemurafenib, encorafenib, and dabrafenib. Vemurafenib was tested in patients positive for the BRAF V600 mutation in the phase I VE-BASKET study. Not all patients with the mutations responded, as the ORR was 33%, while the median PFS was 4.6 months [66]. The GMMG-Birma trial (phase II, *n* = 15) also evaluated a dual inhibition of BRAF (encorafenib) and MEK (binimetinib)—a downstream target of the MAPK pathway—for BRAF-V600-mutated patients with RRMM; the preliminary results showed high efficacy, with an ORR of 82%, and the safety profile was acceptable [67]. There is an opportunity for precision medicine in MM, and several ongoing trials are evaluating MAPK pathway inhibition, whether in monotherapy or in combination with other MM agents.

**Filanesib**. Filanesib is a kinesin spindle protein (KSP) inhibitor that blocks the separation of centrosomes via KSPs during mitosis and induces cell cycle stoppage and apoptosis. KSP inhibition induces the degradation of MCL-1, which results in MM cell death. KSP inhibition is therefore appealing in MM, as the survival of MM cells depends on MCL-1. After it demonstrated activity on MM cells in preliminary models, filanesib was evaluated in a phase I/II study for RRMM either as a single agent or combined with dexamethasone. The phase II part of the trial (single agent *n* = 32; filanesib + dex *n* = 55) led to response rates of 16% (monotherapy) and 15% (with dex) [68]. Filanesib was also evaluated with carfilzomib in a phase I study (64 patients, median of 5 prior lines of therapy)—the ORR was 37% (14% for the K refractory patients), and median PFS was 4.8 months [69]—and with pomalidomide (phase I/II study, *n* = 33, 94% of patients refractory to lenalidomide), with an ORR of 65% and median PFS of 7 months [70]. There are currently no trials ongoing for filanesib in MM, given the other options available, although its association with IMiDs is of interest.

### 4.2. The Biomarker-Driven Strategy

Myeloma’s future could lie in a tailored approach where the treatment strategy can adapted to each patient’s biological and/or molecular features. Theoretically, with the different technologies currently available (NGS, SNP-array, FISH, etc.), it is possible to determine a genomic profile for nearly every patient to not only characterize the prognosis of the disease, but also to identify targets for specific drugs. This precision medicine approach is already used for acute leukemias, and is being investigated for many other hematological conditions.

The ongoing phase I/II MyDRUG (Myeloma Developing Regimens Using Genomics) study (NCT03732703) is testing this tailored approach, as the patients will be enrolled if they present a ≥25% mutation in some specific genes (i.e., *CKDN2C*, *FGR3*, *KRAS*, *NRAS*, *BRAF*
*V600E*, *IDH2*, or *t;11;14*). Patients should have been treated with 1–3 prior lines and been exposed to a PI and an IMiD. The genetic sequencing of MM will be done via another study, MMRF002 (NCT02884102). Afterwards, patients will receive targeted drugs to the mutated genes; some arms of the study include combinations with abemaciclib (CKD inhibitor)–ixazomib–pomalidomide–dex, or cobimetinib (MEK inhibitor)–ixazomib–pomalidomide–dex.

In this way, we could move from a one-size-fits-all treatment strategy to a personalized approach—especially at relapse—in MM. While this might take some time, as it will require further data and phase III clinical trials, the growing knowledge on MM’s genomics and the development of various targeted drugs should allow this manner of treating patients in the years to come.

## 5. Conclusions 

Tremendous progress has been made in the treatment of multiple myeloma over the past few years, especially with the advent of immunotherapy, which has revolutionized the field and contributed greatly to the prolonged survival of patients. However, after periods of remission patients, will eventually relapse, as the disease is still considered to be incurable.

Non-immunologic agents are therefore still relevant in the context of myeloma, particularly in the late clinical course. Indeed, immune-based treatments are mostly used early in the course of myeloma, on naïve plasma cells, when their action will probably be more efficient. However, MM cells will display mechanisms of resistance throughout the course of the disease and, consequently, in late relapses, targeting the immune system might not be the preferred option, as it would have time to adapt beforehand. Moreover, these treatments are nearly exclusively used in combination with other drugs, which makes the options sparse at relapse. For now, modern immunotherapies (i.e., CAR-Ts, bispecifics) are still used in advanced disease, but they will probably be more active in early lines of treatment, as shown by prior investigations, and options with different mechanisms of action are needed. Thus, the treatment of patients with relapsed or refractory multiple myeloma remains challenging, and immunotherapy cannot be the only answer. Moreover, non-immunologic agents demonstrated their efficacy long before the emergence of immunotherapy, and are still the backbone of various treatment combinations.

Targeted treatments such as proteasome inhibitors and immunomodulatory agents have been used in most first-line regimens, but also at relapse, for several decades now. The development of these drugs has continued and led to second- and third-generation drugs, which are possibly more effective, and can often allow the retreatment of patients with the same drug class. However, continuing to develop new generations of IMiDs and PIs might not be the ideal solution, as it will eventually become complicated to find different versions with better efficacy and less toxicity. For example, oprozomib and marizomib—novel-generation PIs—do not seem to bring any additional interest compared to carfilzomib or ixazomib, at least for now. On the other hand, CELMoDs, which represent a different way of targeting the cereblon complex than IMiDs, might find their place in the treatment of myeloma. In this case, CELMoDs cannot really be thought of as new generation IMiDs, as they use a different mechanism of action. Similarly, melflufen—the first peptide conjugate—represents an innovative way to deliver alkylating drugs directly inside MM cells, thus limiting the systemic toxicity encountered with melphalan. This approach is based on preexisting knowledge, but with a modern technology that will likely lead to a revival of alkylating agents.

Nevertheless, for the future, targeting novel pathways is probably the best alternative. Currently, as novel pathophysiological mechanisms are unveiled, entirely new treatments are reinforcing the myeloma armamentarium. Selinexor, the first selective inhibitor of nuclear export, is progressively entering myeloma treatment algorithms, principally for advanced disease, although its safety profile needs to be improved by determining the appropriate dosage. HDAC inhibitors are another option for RRMM, even if their activity as single agent is minimal. Still, they can benefit some patients, especially when combined with PIs or IMiDs. Finally, venetoclax—the first-in-class BCL-2 inhibitor—could become the first biomarker-driven treatment, as it proved very effective for the subgroup of patients harboring the t(11;14) translocation. 

Prospects are bright in the field of myeloma, as novel agents are continuously emerging. Various molecules are in early-stage development, and have shown activity in preclinical models. Different pathways are being explored and targeted, such as the MAPK and JAK pathways, or the inhibition of kinesin spindle proteins and cyclin-dependent kinases. 

The treatment of myeloma is rapidly evolving, but non-immunologic agents should not be abandoned any time soon, as their diversity is still crucial for patients with relapsed or refractory myeloma, and immune-based treatments are still far from providing a curative strategy. 

## Figures and Tables

**Figure 1 cancers-13-05210-f001:**
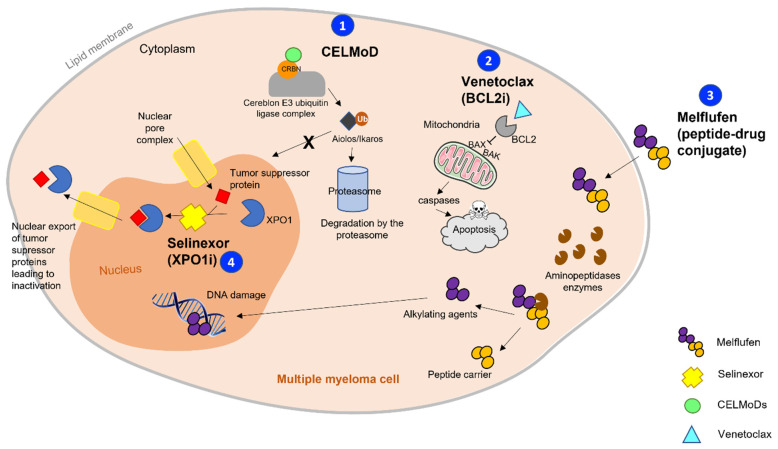
Overview of the mechanisms of action of the most innovative non-immunologic drugs currently in development for multiple myeloma: CELMoDs, BCL-2 inhibitors, peptide–drug conjugates, and XPO-1 inhibitors. (1) CELMoDs bind with high affinity to cereblon—part of the cereblon E3 ubiquitin ligase complex, which modulates the activity of the complex and increases the polyubiquitination of the transcription factors (TFs) Ikaros and Aiolos (zinc finger TFs of the Ikaros family), which are involved in MM cells’ survival and, therefore, their proteasome-dependent degradation. (2) Venetoclax binds to the BH3 domain of BCL-2 and, therefore, frees the pro-apoptotic proteins Bax and Bak that would be inhibited by BCL-2. Bax/Bak activation leads to their mitochondrial outer membrane permeabilization and initiates caspase-mediated apoptosis. (3) Melflufen is a peptide-conjugated alkylator that is rapidly internalized into cells due to its high lipophilicity. As a prodrug, it is metabolized by aminopeptidases, which are increased in MM cells, and selectively releases alkylating agent into tumor cells to exert its cytotoxic activity. (4) Selinexor targets XPO1, which is overexpressed in MM and performs the nuclear export of tumor-suppressor proteins (TSPs). Thus, blocking the action of XPO1 leads to the reactivation of TSPs, and then induces tumor cell apoptosis.

**Table 1 cancers-13-05210-t001:** Available results of the different trials that investigated novel non-immunologic drugs for patients with relapsed or refractory multiple myeloma.

Experimental Drug	Iberdomide (CC-220)	CC-9280	Melflufen	Selinexor	Selinexor
Name of the study	/	/	HORIZON	STORM	BOSTON
NCT/EudraCT number	NCT02773030	NCT03374085	NCT02963493	NCT02336815	NCT03110562
Study phase	I/II	I	II	IIb	III (SVd vs. Vd)
Number of patients	58	66	157	122	402
Median number of prior lines	5	6	5	7	2
Triple-class refractory patients, %	/	50	76	100 Penta-refractory: 68	/
ORR, %	32.2 (CBR = 51)	21 for all evaluable patients, 48 at the therapeutic dose	29 in all evaluable patients, 26 in triple-class refractory	26 CBR: 39	76.4% (for SVd)
Median PFS, months	/	/	4.2	3.7	13.93 (for SVd)
Median OS, months	/	/	11.6	8.6	Not reached

**Table 2 cancers-13-05210-t002:** Ongoing phase III trials investigating the main non-immunologic drugs for relapsed or refractory multiple myeloma.

Name of the Study	EXCALIBER-RRMM	OCEAN	LIGHTHOUSE	CANOVA
Experimental drug	Iberdomide (CC-220)	Melflufen	Melflufen	Venetoclax
NCT/EudraCT number	2020-000431-49	NCT03151811	NCT04649060	NCT03539744
Study arms	Iberdomide–dara–dex vs. DVd	Melflufen–dex vs. poma–dex	Melflufen –dara–dex vs. daratumumab	Venetoclax–dex vs. pomalidomide–dex
Design of the trial	Randomized, controlled, open label	Randomized, controlled, open label	Randomized, controlled, open label	Randomized, open label
Primary endpoint	PFS	PFS	PFS	PFS
Number of prior lines	1–2	2–4	≥3	≥2
Planned number of subjects included	736	450 (Actual enrollment: 495)	240	244

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
