# Peer review of "Novel Non-Immunologic Agents for Relapsed and Refractory Multiple Myeloma: A Review Article"

_cancers, 2021, doi:10.3390/cancers13205210_

Round 1
Reviewer 1 Report
The authors answered all my questions
Reviewer 2 Report
The authors have revised the manuscript according to the comments
This manuscript is a resubmission of an earlier submission. The following is a list of the peer review reports and author responses from that submission.
Round 1
Reviewer 1 Report
The authors provided a comprehensive review on novel non immunologic agents for MM patients. The paper is well written and easy to follow. However, the authors should add a figure summarizing the main mechanism of action of the cited agents
Reviewer 2 Report
The authors present an overview of novel non-immunologic agents for relapse and refractory multiple myeloma. The review article is interesting; however, several aspects need to be addressed:
1) Appropriate references should be added in the introduction
2) Lines 61-62: please add refs and elaborate on myelomas that do not respond to immune therapies
3) Please revise the whole manuscript for minor syntax and grammar errors eg line 84 "combinations" instead of "associations"
4)Line 94 - please comment on the regulatory approved treatment schedule of carfilzomib (twice vs once weekly)
5) Please comment on the role of ixazomib in the maintenance setting, also in view of the recent interim OS analysis of TOURMALINE studies showing an inferior OS compared with control arms
6) Lines 164-165: please elaborate on the safety profile and dose-limiting toxicities of novel CELMODs
7) Lines 169-179: please add appropriate references
8) Melflufen: Please comment on toxicities that led to treatment interruptions/discontinuations and dose modifications (HORIZON trial), and on the FDA warning (https://www.fda.gov/drugs/drug-safety-and-availability/fda-alerts-patients-and-health-care-professionals-about-clinical-trial-results-showing-increased) regarding the increased risk of death in the OCEAN trial
9) The results from BOSTON trial have been published (PMID: 33189178 ) and should be included in Table 1 instead of Table 2
10) Lines 254-255: Please discuss the possible underlying etiology of mortality excess in the experimental arm of the BELLINI trial
11) MCL-1 inhibitors: Please discuss signals of potential toxicity in patients with MM eg. AMG 397
12) Lines 281-288, 331-339: please add references
13) Selinexor: Please discuss toxicity management (PMIDS: 34234609, 32094461)
14) HDAC inhibitors: Can the authors comment on why these agents have not been widely implemented in the clinical practice?
15) section 4.2: The last paragraph on MRD and the comment on the MASTER study (which includes daratumumab-based regimens) is rather general and it may be omitted
16)line 505: I do not think that HDAC inhibitors can be characterized as innovative in 2021
17) Please revise the whole manuscript and add any appropriate references in the text
18) Please add author contributions, since there are 12 authors for a review article, and potential conflict of interests
19) A figure on the non-immunological approaches in RRMM would improve the manuscript
